# Research on the Spatial Dynamic Evolution of Digital Agriculture—Evidence from China

**Jiajia Meng [1], Baoyu Zhao [2,\*], Yuxiao Song [3] and Xiaomei Lin [1]**

1   Sun Wah International Business School, Faculty of Economics, Liaoning University, Shenyang 110036, China; mengjj@lnu.edu.cn (J.M.); linxm0622@163.com (X.L.)
2   School of Economics, Faculty of Economics, Liaoning University, Shenyang 110036, China
3   Department of International Relations, Ritsumeikan University, Kyoto 603-8577, Japan; ir0494ri@ed.ritsumei.ac.jp
\*   Correspondence: 4032230139@smail.lnu.edu.cn; Tel.: +86-187-4532-8235

**Abstract:** Digital agriculture serves as a pivotal means of ushering in innovative agricultural practices and achieving sustainable agricultural development. Although agricultural digitalization has received increasing attention, the unbalanced development and regional disparities of digital agriculture are still key obstacles to sustainable agricultural development. Based on the data of 31 provinces in China from 2013 to 2021, this study evaluates the development level of digital agriculture in China, and further analyzes the distribution pattern, spatial characteristics, and transition probabilities of digital agriculture from a regional perspective. The index system of the digital agriculture development level is constructed from five aspects: infrastructure, talent resources, agricultural informatization, the digitization of agricultural production processes, and agricultural production efficiency. Among these, infrastructure and talent resources reflect the resources needed for the development of digital agriculture; agricultural informatization and the digitization of the agricultural production process indicate the role of digitization in the process of agricultural development; and the agricultural production efficiency is the goal of the digital agriculture development, which is a critical criteria of its evaluation. The weighted analysis method of objective sequential analysis, which combines the dynamic level of indicators and sequential relationships, is used to assign weights to the indicators. In addition, to address the regional disparities in the development level of digital agriculture, kernel density estimation, Moran's index, and (spatial) Markov chain analysis are applied to analyze the spatial dynamic evolution of digital agriculture in China. The findings reveal substantial regional disparities in digital agriculture development within China, particularly in the Western region, where development lags behind. Moreover, this study offers actionable policy recommendations for policymakers to strengthen regional infrastructure and talent cultivation, as well as other aspects of digital agriculture development, to mitigate regional differences and provide reference for other emerging countries.

**Keywords:** digital agriculture; regional differences; objective sequential analysis (SRA) method; sustainable agricultural development

## 1. Introduction

The sustainable development of agriculture, serving as the bedrock of economic progress, is a pivotal prerequisite for achieving high-quality economic growth. Digitalization and informatization represent crucial directions for the future development of agriculture. The rapid proliferation of the digital economy, with its far-reaching influence, acts as a powerful catalyst for the high-quality development of agriculture, offering a novel path towards sustainable agricultural practices.

The concept of digital agriculture first surfaced in 1987, representing the fusion of intensive, information-rich agricultural technology supported by geospatial and information technology [1,2]. Although scholars have not converged on a uniform definition for

"digital agriculture", there is a prevailing consensus that "digital agriculture" signifies the fourth revolution in the field of agriculture [3–5]. The first revolution in agriculture was the transformation from a primitive hunting economy to a farming economy, while the second agricultural revolution turned into mechanical agriculture, which gradually used pesticides, fertilizers, and large machinery in the production process. The third agricultural revolution was labeled as the "Green Revolution", which is mainly reliant on modern biotechnology with molecular biology and genetic engineering, as well as other supporting green technologies that are conducive to sustainable environmental development. The fourth revolution is based on information technology and digitalization, which can more effectively realize the development of precision agriculture [6,7], accelerate the development of agricultural mechanization, and improve agricultural production efficiency [8]. The development of digital agriculture has strengthened the application of the Internet, the Internet of Things [9], cloud computing, and big data to the agricultural production and sales processes, thus continuously improving agricultural production standards, deepening the processing of agricultural products, and building agricultural brands. Therefore, the increase in agricultural output value and sustainable development can be realized [10,11].

Current research places a significant focus on the application of digital technology in the agricultural production process, while also emphasizing efficiency gains, food security, and ecological preservation [12,13]. Research findings highlight the critical role of digitalization in steering agriculture towards sustainable development [14–16]. Digital technologies effectively reduce carbon emissions during production [17,18] and foster green agricultural practices [19]. Digital agriculture enables precise production, enhancing the quality and yield of agricultural products by monitoring dynamic crop changes and effectively regulating and predicting the production process [20–24]. The widespread adoption of digital machinery and equipment reduces fixed agricultural operation costs [25], while increasing overall agricultural productivity [26,27]. This shift towards reduced energy consumption and heightened productivity propels agriculture towards sustainability [10,11,28]. The confluence of the "digital economy" and "agriculture" expands avenues for agricultural sales [29], enhances market transparency [30], augments agricultural production efficiency [31,32], balances supply and demand information, and delivers high-quality agricultural products to consumers. Moreover, digital agriculture stimulates the growth of agriculture-related industries [33], elevates agricultural and supply chains [34,35], and reinvigorates rural sectors [36].

As an important developing country in the agricultural field, China has experienced a rapid growth in digital agriculture in recent years, with achievements such as the informatization level of national agricultural production reaching 22.5%, agricultural product quality and safety traceability at 22.1%, and online retail sales of agricultural products accounting for 13.8% of total agricultural product sales in 2020 [37], which make China become an important force for the sustainable development of agriculture in the world. The significant role of China in global agriculture and digital agriculture is remarkable, such as its support of nearly one-fifth of the world's population with less than 9 percent of the world's total arable land [38]; furthermore, China is also exporting the experience of agricultural digitization to developing countries.

However, when considering the characteristics of the agricultural industry and the disparities in economic development levels, it becomes apparent that the degree of digital transformation in Chinese agriculture exhibits noticeable imbalances [39] and that the progress of digital agriculture in China also shows obvious regional disparities. Existing studies have revealed the challenges posed by these imbalances in digital agriculture development, and the substantial regional disparities represent a significant hurdle to coordinated and sustainable agricultural development [40]. Consequently, the issue of the regional divide in digital economy development has become a pressing concern [41,42].

To comprehensively analyze regional disparities in digital agriculture development in China and propose actionable policy recommendations, this paper constructs an evaluation index system for the digital agriculture development level, encompassing multiple dimen-

sions. The study also employs objective sequential analysis to assess the development level of digital agriculture across 31 provinces in China. Additionally, this research delves into the distribution patterns, spatial characteristics, and transition probabilities of digital agriculture development levels within the three major regions, providing a detailed portrait of regional differences in digital agriculture across spatial and temporal dimensions. This study enriches the understanding of digital agriculture development as well as regional disparities, while offering actionable policy recommendations to mitigate regional differences in digital agriculture development and providing reference for other emerging countries.

## 2. Index System and Methodology

### 2.1. Index System of Digital Agriculture Development Level

The progression of digital agriculture development is intricate, and a solitary indicator falls short of providing an objective evaluation. Consequently, most scholars rely on the construction of an index system to gauge the development level of digital agriculture [43–45]. In this paper, we have formulated an evaluation system based on insights from the "2021 National County Agricultural and Rural Informatization Development Level Evaluation Report, (in China)" and the "Digital Countryside Development Report in China", as well as other pertinent research reports and studies. This comprehensive system comprises five core dimensions and encompasses 18 key indicators, which include digital agriculture infrastructure, digital agriculture talent, the agricultural informatization level, the digitization of the agricultural production process, and the agricultural production efficiency, as shown in Table 1.

**Table 1.** Evaluation system of digital agriculture development level.

| Dimension | Indicator | Unit | Code | Property |
|---|---|---|---|---|
| Digital agriculture infrastructure | Rural delivery routes | Kilometer | $X_1$ | Benefit |
| | Total reservoir capacity | Billion cubic meters | $X_2$ | Benefit |
| | Road mileage | Kilometer | $X_3$ | Benefit |
| | Rural electricity consumption | 10,000 kW | $X_4$ | Benefit |
| | Length of optical fiber cables | Kilometer | $X_5$ | Benefit |
| Digital agriculture talent resources | Average years of education in rural areas | Year | $X_6$ | Benefit |
| | Education expenditure | 10,000 yuan | $X_7$ | Benefit |
| | Science and technology expenditure | 10,000 yuan | $X_8$ | Benefit |
| | Average number of students enrolled in higher education per 100,000 population | Number | $X_9$ | Benefit |
| Agricultural informatization level | The density of cell phone base stations | 10,000 $km^2$ | $X_{10}$ | Benefit |
| | The number of Internet domain names per 1000 people | 10,000 | $X_{11}$ | Benefit |
| | Rural broadband access users | 10,000 | $X_{12}$ | Benefit |
| Digitization of agricultural production process | The effective irrigated area | Thousand hectares | $X_{13}$ | Benefit |
| | The total power of agricultural machinery | 10,000 kW | $X_{14}$ | Benefit |
| | Large and medium-sized tractors for agricultural use | Number | $X_{15}$ | Benefit |
| Agricultural production efficiency | E-commerce sales | 100,000,000 yuan | $X_{16}$ | Benefit |
| | Per capita disposable income of rural households | Yuan | $X_{17}$ | Benefit |
| | The total output value of agriculture, forestry, animal husbandry, and fisheries | 100,000,000 yuan | $X_{18}$ | Benefit |

Infrastructure forms the bedrock for digital agriculture development, which encompasses five crucial aspects: rural delivery routes, total reservoir capacity, road mileage, rural electricity consumption, and the length of optical fiber cables. Rural delivery routes and road mileage represent the transportation infrastructure essential for the movement of agricultural production materials and products. Meanwhile, total reservoir capacity and rural electricity consumption are fundamental resources vital for digitizing the agricultural production process. The length of optical fiber cables serves as the hardware foundation underpinning the advancement of agricultural informatization and digitization.

Talent resources play an indispensable role in elevating the level of digital agriculture development. In comparison to hardware infrastructure, the scarcity of human resources

remains a primary challenge in the digital agriculture development process and stands as a pivotal indicator reflecting the current state of digital agriculture development. Talent resources within digital agriculture are evaluated through four key indicators: average years of education in rural areas, education expenditure, science and technology expenditure, and the average number of students enrolled in higher education per 100,000 population.

The level of agricultural informatization is a critical dimension that mirrors the state of digital agriculture development. The application of digital technology has made data and information pivotal in agricultural progress. Evaluation indicators for the level of agricultural informatization encompass the density of cell phone base stations, the number of Internet domain names per 1000 people, and the count of rural broadband access users.

The digitization of the agricultural production process represents a key facet of digital agriculture development. The essential roles of digital agriculture are to refine the production process, enhance agricultural production efficiency, and conserve resources. This dimension includes indicators such as the effective irrigated area, the total power of agricultural machinery, and the number of large and medium-sized tractors used in agriculture. These indicators provide insights into the capacity and degree of digitization of the production process in various regions.

The agricultural production efficiency serves as the ultimate objective of digital agriculture development. Vigorously promoting the digital agriculture process aims to achieve high-quality agricultural development and sustainability, with the ultimate benefits reflected in the efficiency of agricultural production. This dimension includes indicators such as e-commerce sales, per capita disposable income of rural households, and the total output value of agriculture, forestry, animal husbandry, and fisheries.

### 2.2. Methods of Measuring the Development Level of Digital Agriculture

In the contemporary landscape of digital agriculture development assessment, traditional methods such as the entropy value method and principal component analysis are widely employed. However, these conventional approaches tend to underemphasize the inherent significance of individual indicators. In light of this, this paper adopts a novel weighted analysis technique that integrates the dynamic evolution and sequential relationships of indicators, referred to as objective sequential analysis (SRA). SRA represents a weighted analysis approach that intricately blends the dynamic evolution and sequential relationships among indicators. This method places heightened importance on discerning the relative significance of indicators within the evaluation framework. It achieves this by scrutinizing the variations in indicators across different evaluation periods, thereby objectively ascertaining their weights within the evaluation system.

Consider a scenario where there are $n$ alternatives, denoted as $A_1, A_2, \cdots, A_n$, measured by $m$ indicators, marked as $X_1, X_2, \cdots, X_m$. Without a loss of generality, let $a_{ij}(t_k)$ represent the actual performance of alternative $A_i$ on indicator $X_j$ in assessment period $t_k$, where $k = 1, 2, \cdots, T$ represents $T$ periods. In the objective sequential analysis (SRA) method, the comparative evaluation of indicators is quantified based on their competitive capability, rather than relying on subjective judgments from experts. Consequently, the initial step in this process involves determining the competitive capabilities of different indicators by analyzing the positive shifts in their ranking values.

Let $\lambda_j$ represents the competitive capability of indicator $X_j$, then:

$$\lambda_j = \frac{count\big(\big(r_{ij}(t_{k+1}) - r_{ij}(t_k)\big) < 0\big)}{n(T-1)}, i = 1, 2, \cdots, n; j = 1, 2, \cdots, m; k = 1, 2, \cdots, T-1 \tag{1}$$

where $r_{ij}(t_k)$ indicates the ranking value of alternative $A_i$ out of all the alternatives on indicator $X_j$ in period $t_k$, with the values ranged in $[1, n]$. $count(\cdot)$ is a counting function used to count the number of sort improvements in the next period over the adjacent previous period of alternative $A_i$ on indicator $X_j$.

It is known that $count\big(\big(r_{ij}(t_{k+1}) - r_{ij}(t_k)\big) < 0\big) \in [0, n(T-1)]$. Therefore, $\lambda_j \in [0, 1]$. A larger value of $\lambda_j$ indicates the higher competitive capability of indicator $X_j$.

Based on the analysis above, the objective SRA method is proposed as follows:

Step 1—Rearrange indicators by their competitive capability.

Rearrange the indicators by their competitive capability $\lambda_j$. An indicator with a higher competitive capability (larger value of $\lambda_j$) ranks on the top of the ordered indicators. For simplicity, we denote the ordered indicators as $X_1 \succ X_2 \succ \cdots \succ X_m$, where "$\succ$" represents "proceed".

Step 2—Determine the ratio of the importance of two adjacent indicators.

The ratio of importance of two adjacent indicators $X_j$ and $X_{j+1}$, denoted as $r_j$, can be determined by:

$$r_j = \frac{\lambda_{j-1}}{\lambda_j}, \, j = 2, \ldots\ldots, m-1 \tag{2}$$

Step 3—Calculate the indicator weights

Let variable $w_j$ be the weight of indicator $X_j$. Then, we have:

$$\begin{cases} w_m = \left(1 + \sum_{l=2}^m \prod_{j=l}^m r_j\right)^{-1} \\ w_{l-1} = r_l w_l, l = m, m-1, \cdots, 2 \end{cases} \tag{3}$$

The validity of Equation (3) is proved in the following.

**Proof.** Since $w_{j-1}/w_j = r_j$ for $j = 2, \cdots, m-1$, we have $\prod_{j=2}^m r_j = w_1/w_m$.

Then, $\sum_{l=2}^m \prod_{j=l}^m r_j = \frac{\sum_{j=1}^{m-1} w_j}{w_m}$.

Therefore, $1 + \sum_{l=2}^m \prod_{j=l}^m r_j = 1 + \frac{\sum_{j=1}^{m-1} w_j}{w_m} = \frac{\sum_{j=1}^m w_j}{w_m}$.

Since $\sum_{j=1}^m w_j = 1$, we have $w_m = \left(1 + \sum_{l=2}^m \prod_{j=l}^m r_j\right)^{-1}$.

The equation $w_{l-1} = r_l w_l, l = m, m-1, \cdots, 2$ can be deduced from $w_{j-1}/w_j = r_j$. $\square$

Once the weights for each indicator have been determined, the next step involves calculating the evaluation values for the five dimensions: digital agriculture infrastructure, digital agriculture talent resources, the agricultural informatization level, the digitization of the agricultural production process, and agricultural production efficiency. These evaluation values for each dimension are represented as $L_{ic}$, $L_{tr}$, $L_{it}$, $L_d$, and $L_b$, and the level of digital agriculture development is represented by $L$. Where $i$ represents provinces and $j$ represents indicators, the specific formula expressions are as follows:

$$L_{iic}(t_k) = \sum_{j=1}^5 w_j y_{ij}(t_k), i = 1, 2, \cdots, n; j = 1, 2, \cdots, 5 \tag{4}$$

$$L_{itr}(t_k) = \sum_{j=6}^9 w_j y_{ij}(t_k), i = 1, 2, \cdots, n; j = 6, 7, \cdots, 9 \tag{5}$$

$$L_{iit}(t_k) = \sum_{j=10}^{12} w_j y_{ij}(t_k), i = 1, 2, \cdots, n; j = 10, 11, 12 \tag{6}$$

$$L_{id}(t_k) = \sum_{j=13}^{15} w_j y_{ij}(t_k), i = 1, 2, \cdots, n; j = 13, 14, 15 \tag{7}$$

$$L_{ib}(t_k) = \sum_{j=16}^{18} w_j y_{ij}(t_k), i = 1, 2, \cdots, n; j = 16, 17, 18 \tag{8}$$

$$L_i(t_k) = (L_{ic} + L_{tr} + L_{it} + L_d + L_b)/5 \tag{9}$$

In order to ensure the dynamic changes in different evaluation periods, the data are standardized using the method of dynamic extreme value. As all the indicators in the evaluation index system of digital agriculture development level are positive indicators, their processed values are set as $y_{ij}(t_k)$, and the formula is as shown in Equation (10):

$$y_{ij}(t_k) = \frac{x_{ij}(t_k) - \min(x_{ij}(t_k))}{\max(x_{ij}(t_k)) - \min(x_{ij}(t_k))} \tag{10}$$

### 2.3. Methods for Assessing Spatial and Temporal Trends in the Development Level of Digital Agriculture

(1) Distribution pattern—Kernel density estimation.

Kernel density estimation is one of the methods of spatial disequilibrium analysis. The position of the kernel density curve is utilized to reflect the level of digital agriculture development. We utilize the height and width of wave peaks to illustrate the degree of discrete agglomeration. The number of wave peaks is used to illustrate the polarization phenomenon, and the ductility-dragging tail phenomenon is used to illustrate the degree of divergence.

It is assumed that the density function of the random variable $x$ of the method is as follows:

$$f(x) = \frac{1}{Nh}\sum_{i=1}^{N} K\left(\frac{X_i - x}{h}\right) \tag{11}$$

where $N$ is the number of observations; $h$ is the bandwidth; $K$ is the kernel density; $X_i$ is the independent identically distributed observations; and $x$ is the mean value. In addition, it is necessary to satisfy: $\lim_{x\to\infty} K(x)\cdot x = 0$; $K(x) \geq 0$; $\int_{-\infty}^{+\infty} K(x)dx = 1$; $supK(x) < +\infty$; $\int_{-\infty}^{+\infty} K^2(x)dx < +\infty$.

(2) Spatial autocorrelation—Moran index and Lisa plot.

Whether the digital agriculture development level has spatial autocorrelation needs to be tested by autocorrelation. Global spatial autocorrelation is mainly used to measure whether there are agglomeration characteristics of spatial units, while local spatial autocorrelation explains the agglomeration characteristics of specific spatial locations and the significance of agglomeration, which is presented by Lisa diagrams. The global Moran's $I$ and local Moran's $I$ are specifically represented as follows:

$$I_G = \sum_{i=1}^{n} \sum_{j=1}^{n} w_{ij}(y_i - \overline{y})(y_j - \overline{y}) / S_0 S^2 \tag{12}$$

$$I_L = Z_i \sum_{j\neq i}^{n} w_{ij} Z_j / S^2 \tag{13}$$

where $S_0 = \sum_{i=1}^{n}\sum_{j=1}^{n} w_{ij}$, $S^2 = \sum(y_i - \overline{y})^2/n$, $Z_i = y_i - \overline{y}$, $Z_j = y_j - \overline{y}$, $n$ is the number of spatial units, $y_i$ and $y_j$ are the attribute values of the ith and jth spatial units, $w_{ij}$ is the spatial weight value, using spatial neighbor weights, and $\overline{y}$ is the average of the attribute values of all spatial units.

(3) Evolutionary dynamics—Markov chains.

Markov chains are utilized to illustrate the transfer probabilities of the digital agriculture development level at different states. The Markov chain is a $k \times k$ matrix of transfer probabilities depicting the transfer of attribute types at different times, with the probability of a region of type $i$ at time t transferring to type $j$ at the next moment denoted by $a_{ij}$:

$$a_{ij} = b_{ij}/b_i \tag{14}$$

Assuming that the Markov transfer probabilities are smooth in time, we have the following:

$$N_{t+s} = M^S N_t \tag{15}$$

where $b_{ij}$ denotes the number of areas transferred from type $i$ to type $j$ in the study time period, from time period t to the next time period, and $b_i$ denotes the number of areas of type $i$ in time period $t$. $M^s$ is the $s$th power of the transfer probability matrix $M$, and $N_t$, $N_{t+s}$ are the probability distributions over time periods $t$ and $t + s$, respectively.

While the spatial Markov chain method is formed by introducing spatial autocorrelation into the Markov chain method, the transfer probability matrix is conditioned on the type of spatial lag in a region at time $t$. The spatial lag value of region $u$ is a weighted

average of the values of attributes of spatially neighboring regions of the region, which is given in the following formula:

$$Lag = \sum_{u=1}^{U} y_u w_{uv} \qquad (16)$$

where the spatial weight matrix $w_{uv}$ denotes the spatial relationship between region $u$ and region $v$ using the spatial adjacency matrix, $y_u$ denotes the attribute value of region $b$, and *Lag* is the spatial lag value of region $u$, which denotes the state of the region $u$ neighborhood.

## 3. Results

### 3.1. Results of Measuring the Development Level of Digital Agriculture

Based on the objective sequential analysis (SRA), the weight of each indicator of the digital agriculture development level is calculated, and the results are shown in Table 2.

**Table 2.** Weights of indicators calculated based on objective sequential analysis.

| **Digital Agriculture Infrastructure** | $X_1$ 0.235 | $X_2$ 0.128 | $X_3$ 0.208 | $X_4$ 0.155 | $X_5$ 0.274 |
|---|---|---|---|---|---|
| Digital agriculture talent resources | $X_6$ 0.309 | $X_7$ 0.254 | $X_8$ 0.158 | $X_9$ 0.279 | |
| Agricultural informatization level | $X_{10}$ 0.152 | $X_{11}$ 0.481 | $X_{12}$ 0.367 | | |
| Digitization of agricultural production process | $X_{13}$ 0.385 | $X_{14}$ 0.231 | $X_{15}$ 0.385 | | |
| Agricultural production efficiency | $X_{16}$ 0.280 | $X_{17}$ 0.268 | $X_{18}$ 0.452 | | |

The assessment of digital agriculture development across 31 provinces in China over the period 2013–2021 has yielded valuable insights into regional disparities and progress. The specific data involved in the process of measuring the "digital agriculture development level" come from the China Statistical Yearbook, provincial statistical yearbooks, statistical bulletins, etc. There are a few missing values in some indicators; therefore, this study uses linear interpolation to supplement the corresponding indicators to ensure the completeness of the data.

The evaluation results, presented in Table 3, highlight the varying levels of digital agriculture development in these regions. The top three regions with the highest level of digital agriculture development are Shandong Province, Jiangsu Province, and Guangdong Province, while the bottom three regions are Ningxia Autonomous Region, Qinghai Province, and Tibet Autonomous Region.

By examining the average development level and rankings of digital agriculture across each province, it becomes evident that the eastern and central regions generally boast better digital agriculture development levels. In contrast, the western region lags behind, experiencing a more substantial gap compared to other regions. Overall, the development level of digital agriculture in China has been continuously rising. However, regional disparities still persist.

To gain a better understanding of the digital agriculture development level and regional disparities among Chinese provinces, a heat map depicting the level of digital agriculture development in China was generated, based on the average values obtained from Table 3. The heat map utilizes a color spectrum, with darker shades indicating higher levels of digital agriculture development and lighter shades representing lower levels. As depicted in Figure 1, a gradual transition from darker to lighter colors is observed from east to west, reflecting a better overall development of digital agriculture in the eastern and central regions compared to the western region. The disparity between the eastern and central regions is relatively minimal, which possibly due to the fact that agricultural devel-

opment is intrinsically linked to the natural environment, and the central region's provinces possess a stronger foundation for digital agriculture implementation. Additionally, the eastern region benefits from developed economic conditions and access to coastal areas and abundant water resources, which contribute to its digital agriculture advancement. Consequently, both the eastern and central regions exhibit relatively favorable progress in digital agriculture development.

**Table 3.** Evaluation of digital agriculture development in 31 provinces in China (2013–2021).

| Province | 2013 | 2015 | 2017 | 2018 | 2019 | 2020 | 2021 | Average | Rank |
|---|---|---|---|---|---|---|---|---|---|
| Shandong | **43.41** | **46.77** | **47.85** | **50.17** | **51.66** | **54.16** | **57.69** | **49.10** | **1** |
| Jiangsu | **36.10** | **40.37** | **44.95** | **47.27** | **50.57** | **52.97** | **52.30** | **44.94** | **2** |
| Guangdong | **31.46** | **36.58** | **42.02** | **45.38** | **51.01** | **52.14** | **53.86** | **42.74** | **3** |
| Henan | 36.52 | 40.21 | 41.13 | 43.29 | 45.15 | 47.24 | 49.41 | 42.30 | 4 |
| Hebei | 34.31 | 36.01 | 35.48 | 37.48 | 39.54 | 41.12 | 42.45 | 37.19 | 5 |
| Sichuan | 23.86 | 28.62 | 33.29 | 36.21 | 39.59 | 42.37 | 44.67 | 33.87 | 6 |
| Hunan | 26.31 | 28.81 | 32.65 | 34.25 | 37.11 | 38.79 | 40.54 | 32.92 | 7 |
| Hubei | 26.91 | 30.08 | 32.13 | 33.89 | 36.47 | 38.03 | 39.82 | 32.90 | 8 |
| Zhejiang | 25.37 | 28.96 | 31.86 | 34.82 | 38.20 | 38.83 | 39.63 | 32.71 | 9 |
| Anhui | 25.00 | 28.38 | 31.31 | 33.95 | 36.32 | 38.31 | 40.92 | 32.33 | 10 |
| Beijing | 20.78 | 27.14 | 31.42 | 30.42 | 33.53 | 33.11 | 38.09 | 29.75 | 11 |
| Heilongjiang | 24.32 | 25.79 | 27.49 | 28.68 | 30.37 | 31.38 | 32.19 | 27.87 | 12 |
| Fujian | 16.83 | 19.87 | 27.05 | 27.48 | 29.63 | 27.04 | 29.23 | 24.20 | 13 |
| Shanghai | 15.14 | 21.50 | 24.95 | 25.35 | 27.43 | 29.15 | 29.97 | 24.10 | 14 |
| Guangxi | 17.74 | 19.96 | 22.81 | 24.88 | 27.95 | 29.62 | 31.43 | 23.83 | 15 |
| Yunnan | 17.25 | 20.26 | 22.02 | 23.56 | 26.40 | 28.25 | 29.64 | 22.95 | 16 |
| Jiangxi | 16.93 | 19.02 | 22.06 | 23.89 | 26.51 | 28.21 | 29.63 | 22.65 | 17 |
| Liaoning | 19.84 | 21.68 | 21.93 | 22.75 | 23.42 | 24.37 | 25.73 | 22.41 | 18 |
| Shaanxi | 18.13 | 19.64 | 20.57 | 22.09 | 24.12 | 25.00 | 25.91 | 21.58 | 19 |
| Inner Mongolia | 17.70 | 19.23 | 20.41 | 21.56 | 23.23 | 24.35 | 25.57 | 21.08 | 20 |
| Xinjiang | 16.34 | 18.42 | 19.96 | 21.68 | 22.71 | 23.78 | 25.64 | 20.55 | 21 |
| Jilin | 16.68 | 18.33 | 18.43 | 19.97 | 21.39 | 22.80 | 23.87 | 19.67 | 22 |
| Shanxi | 16.78 | 18.12 | 17.17 | 18.74 | 19.29 | 20.15 | 21.41 | 18.41 | 23 |
| Guizhou | 12.26 | 14.43 | 17.02 | 19.07 | 20.74 | 21.93 | 24.33 | 17.64 | 24 |
| Chongqing | 12.31 | 14.62 | 16.84 | 17.82 | 19.81 | 21.18 | 22.56 | 17.17 | 25 |
| Tianjin | 13.20 | 14.96 | 15.03 | 15.78 | 17.04 | 17.64 | 19.23 | 15.77 | 26 |
| Gansu | 13.31 | 14.65 | 14.68 | 16.10 | 17.25 | 18.10 | 19.18 | 15.72 | 27 |
| Hainan | 8.95 | 10.30 | 11.94 | 12.32 | 13.95 | 14.16 | 14.35 | 11.76 | 28 |
| Ningxia | **6.84** | **7.54** | **7.92** | **8.60** | **9.45** | **10.09** | **10.54** | **8.44** | **29** |
| Qinghai | **4.50** | **5.47** | **6.80** | **7.43** | **7.74** | **8.08** | **8.56** | **6.65** | **30** |
| Tibet | **3.74** | **5.02** | **5.70** | **6.06** | **6.54** | **7.29** | **8.04** | **5.74** | **31** |

The division of Chinese provinces into high and low development levels of digital agriculture, based on average values, offers a clear picture of the regional disparities in digital agriculture advancement. The results are depicted in Figure 2. There are 12 provinces classified as having a high development level of digital agriculture, while 19 provinces fall into the low-level category. This result indicates that approximately 60% of China's regional digital agriculture development still requires further improvement. Among the twelve provinces with a high development level of digital agriculture, six of them are located in the eastern region, namely Shandong, Jiangsu, Guangdong, Hebei, Zhejiang, and Beijing. The central region encompasses five provinces, namely Henan, Hunan, Hubei, Anhui, and Heilongjiang. Only Sichuan represents the western region as a province with a high development level of digital agriculture. The distribution of high-level digital agriculture provinces once again underscores the significant regional disparities in China's digital agriculture development. This has emerged as one of the critical challenges impeding the sustainable development of Chinese agriculture.

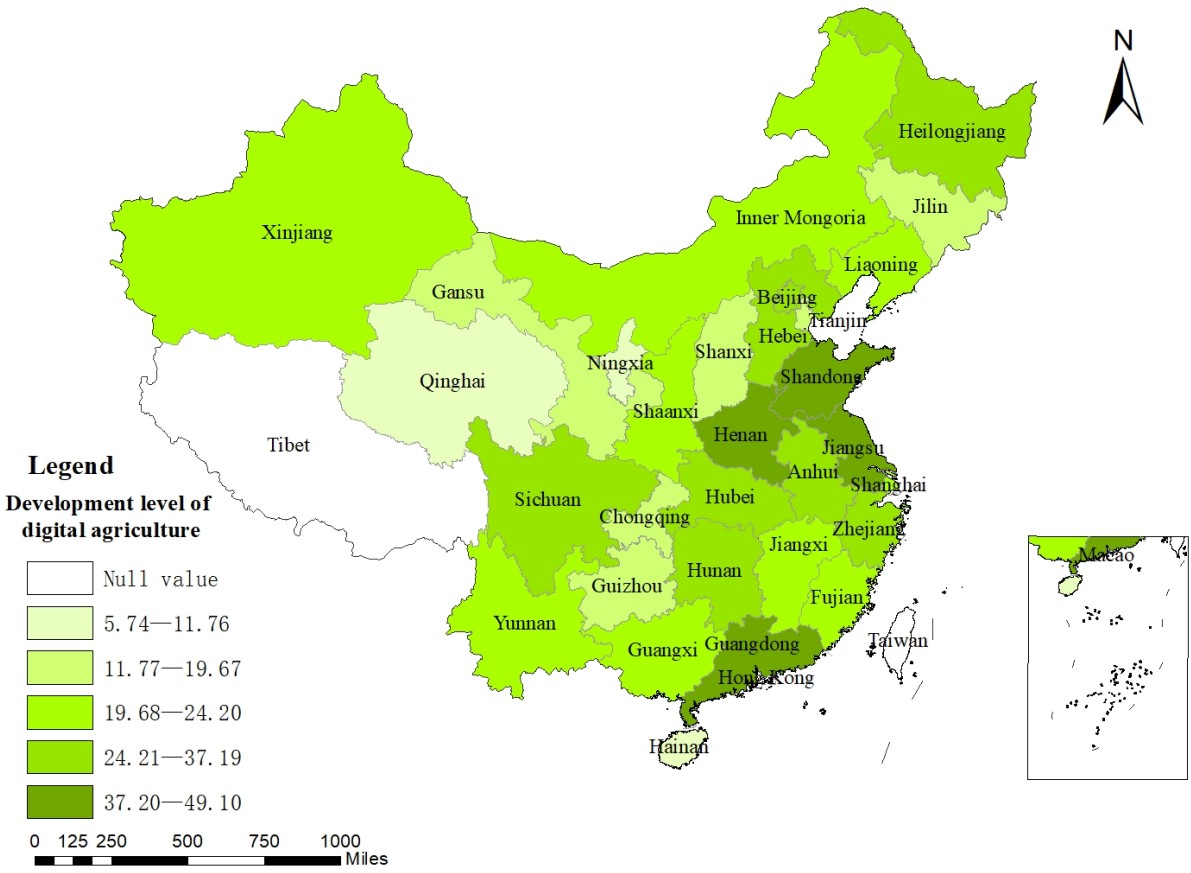

**Figure 1.** Heat map of digital agriculture development level in China.

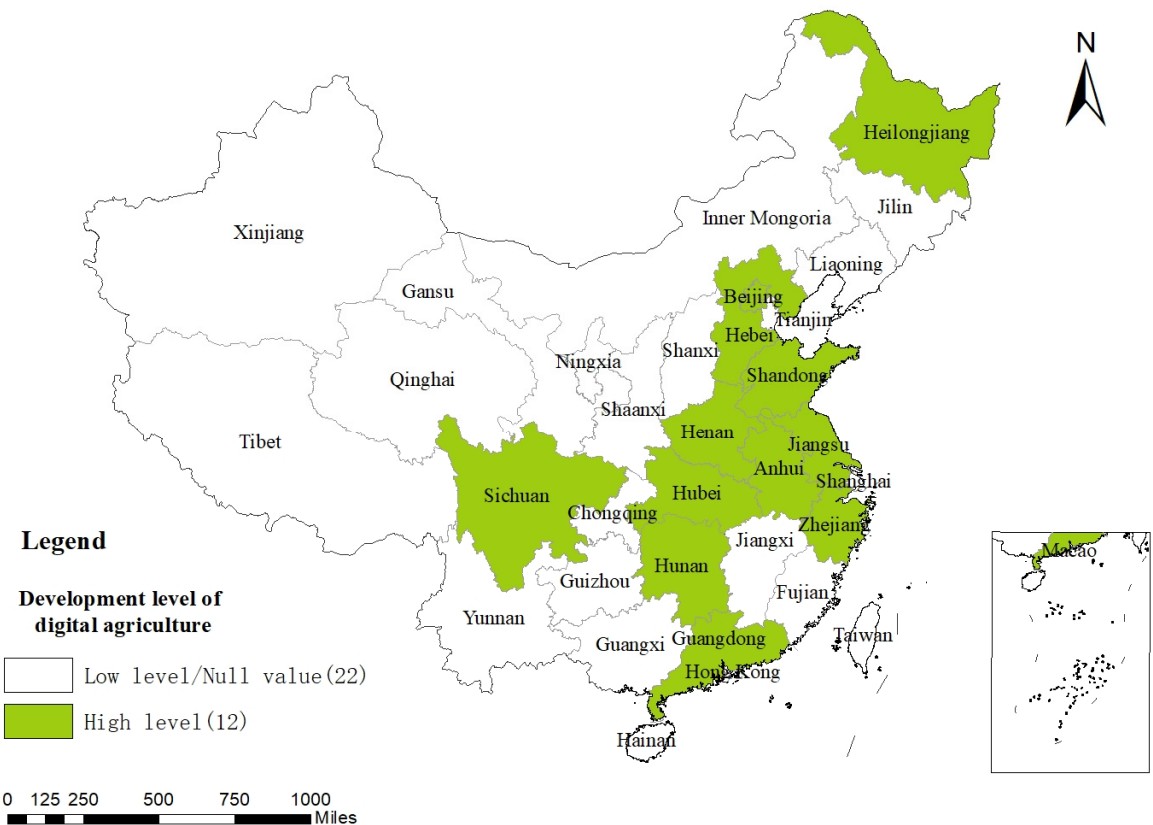

**Figure 2.** Regional distribution map of digital agriculture level in China.

### *3.2. Distribution Pattern*

In order to observe the distribution pattern of the digital agriculture development in each region of China more clearly, a three-dimensional kernel density map of the digital agriculture development is shown in Figure 3, including the eastern, central, and western regions, as well as the whole country, from 2013 to 2021.

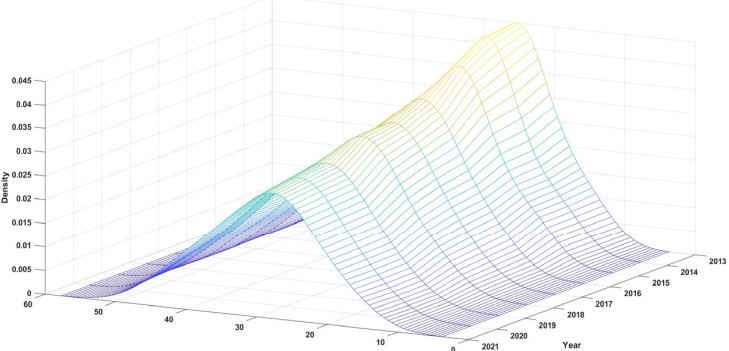

National Development Level of Digital Agriculture

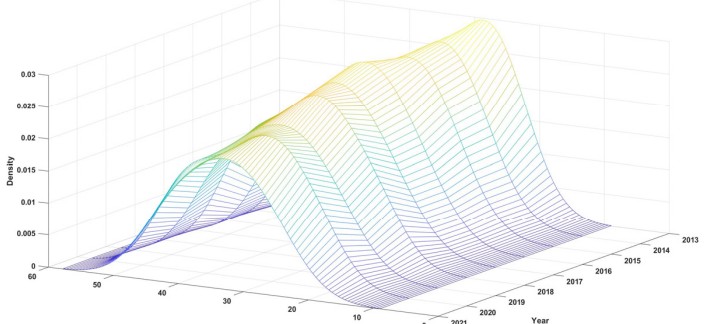

The Development Level of Digital Agriculture in the Western Region

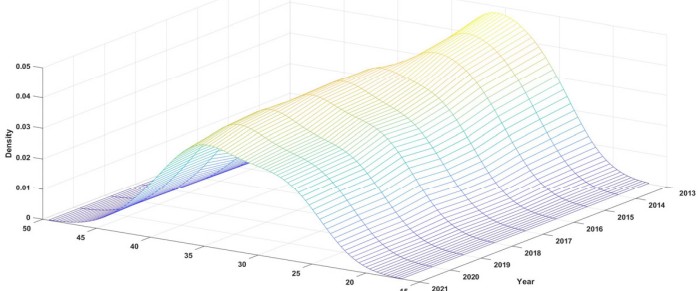

The Development Level of Digital Agriculture in the Central Region

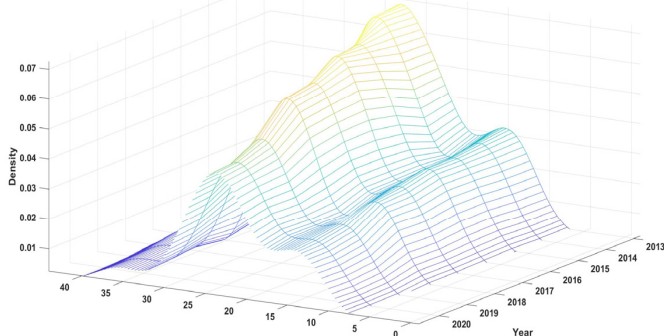

The Development Level of Digital Agriculture in the Western Region

**Figure 3.** Kernel density estimation maps.

The analysis of the distribution pattern of the regional digital agriculture development level in China provides several noteworthy conclusions.

Firstly, across the nation and across the eastern and central regions, there are observable leftward shifts in the center of the kernel density curves. This indicates that the development level of digital agriculture has improved to varying degrees in these regions. However, there is no clear leftward shift trend evident in the western region.

Secondly, the primary peaks in the kernel density curves for the nation as a whole and the eastern, central, and western regions have experienced varying degrees of decrease. Additionally, the width of the wave peaks in the eastern and central regions is increasing, indicating a widening gap in the development level of digital agriculture within these regions.

Finally, in all regions except the western region, the kernel density curves consistently exhibit single peaks. This implies that the development level of digital agriculture in each province within these regions is relatively consistent. In contrast, the western region maintains a double-peak trend from 2013 to 2021, indicating a "polarization" within the region. While Sichuan, as a large agricultural province, has achieved higher levels of digital agriculture development, due to favorable natural conditions and economic resources, many other provinces in the western region still lag behind. This significant gap in development within the western region contributes to the presence of double peaks.

### 3.3. Spatial Autocorrelation

(1) Global Autocorrelation

By utilizing the Local Moran's *I* function in GeoDa 10.8 software, the global Moran's *I* index was calculated, and the results are presented in Table 4. Analyzing the Moran's *I* values, it can be observed that they initially increase and then decrease over time. Although there is a declining trend in the Moran's *I* values from 2016 to 2021, they still remain above 0.260. This suggests that there is an ongoing positive correlation in the development level of digital agriculture. Furthermore, considering the level of significance, the maximum value of 0.11 during the period from 2013 to 2021 meets the requirement for significance. This indicates the presence of a positive spatial dependence in the pattern of digital agriculture development in China, along with a spatial aggregation phenomenon.

**Table 4.** Global Moran's *I* of digital agriculture development level (2013–2021).

|  | **2013** | **2014** | **2015** | **2016** | **2017** | **2018** | **2019** | **2020** | **2021** |
|---|---|---|---|---|---|---|---|---|---|
| Moran's *I* | 0.296 | 0.311 | 0.303 | 0.306 | 0.298 | 0.289 | 0.279 | 0.270 | 0.269 |
| z-value | 2.816 | 2.952 | 2.889 | 2.919 | 2.852 | 2.768 | 2.694 | 2.608 | 2.599 |
| *p*-value | 0.005 | 0.005 | 0.005 | 0.005 | 0.007 | 0.008 | 0.010 | 0.009 | 0.011 |

(2) Local spatial autocorrelation

The global Moran's *I* results reflect the existence of a spatial aggregation relationship of China's digital agriculture development level. In order to better analyze the characteristics of the clustering of digital agriculture development level in each province, the local Moran's *I* index is calculated, and the significant Lisa clustering map of China's digital agriculture development level in 2013 and 2021 is drawn by Arcgis, as shown in Figure 4.

In 2015, the clustering of six provinces is significant, including three high–high clustering areas: Shandong, Henan, Anhui; two low–low clustering areas: Gansu and Xinjiang; and one high–low clustering area: Sichuan. In 2019, the clustering of eight provinces is significant, including less Henan and more Jiangsu in the high–high clustering area, more Inner Mongolia in the low–low clustering area; still only Sichuan in the high–low clustering area; and Jiangxi appears in the low–high clustering area. The results of the significant agglomerations to which each province belongs are shown in Table 5.

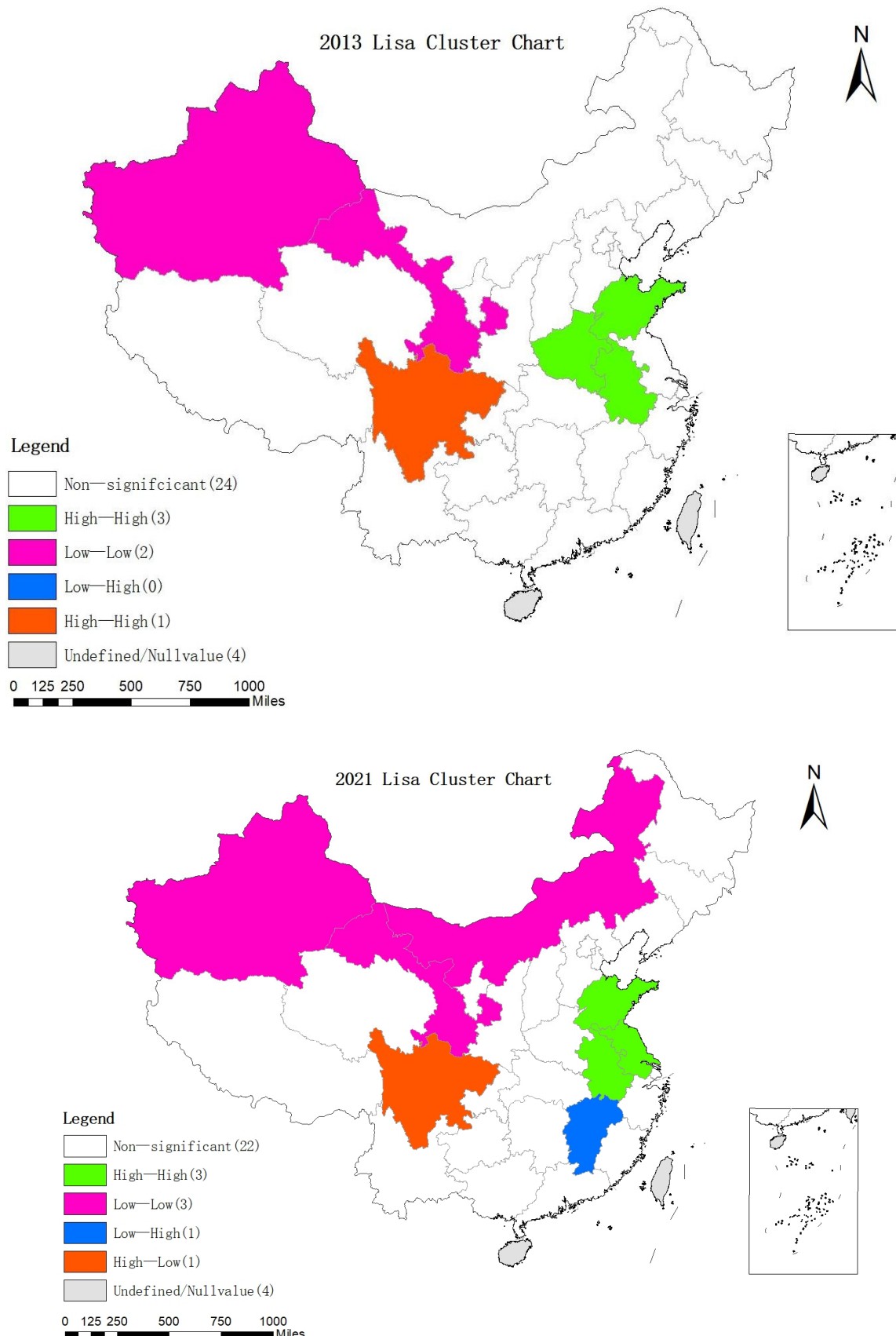

**Figure 4.** Lisa clustering in 2013 and 2021.

**Table 5.** Significant clustering of the digital agriculture development by province in 2013, 2017, and 2019.

| Year | High–High Clustering Areas | High–Low Clustering Area | Low–Low Clustering Area | Low–High Clustering Area |
|------|---------------------------|-------------------------|------------------------|-------------------------|
| 2013 | Shandong, Henan, Anhui | Sichuan | Xinjiang, Gansu | |
| 2017 | Shandong, Jiangsu, Anhui | Sichuan | Xinjiang, Gansu | Jiangxi |
| 2021 | Shandong, Jiangsu, Anhui | Sichuan | Xinjiang, Inner Mongolia, Gansu | Jiangxi |

Based on the findings presented in Table 5, the following conclusions can be derived.

Firstly, it is evident that there is a spatial clustering pattern in the level of digital agriculture development in China, and this clustering tendency is progressively strengthening.

Secondly, there is a tendency for enterprises with similar levels of digital agriculture development to cluster spatially. This means that provinces with high development levels of digital agriculture are more likely to be surrounded by other provinces with high levels of development, while provinces with low development levels tend to be surrounded by other provinces with similarly low development levels.

Lastly, by examining the quantity of high–high cluster areas, low–low cluster areas, high–low cluster areas, and low–high cluster areas, we can infer that the regional disparities in digital agriculture development within the country are relatively minimal.

*3.4. Transfer Probability*

(1) Transfer probabilities for the development level of digital agriculture.

Utilizing the quartile principle, the development level of digital agriculture is categorized into four groups: low level, medium–low level, medium –high level, and high level. By evaluating the development level of digital agriculture in 31 provinces from 2013 to 2021, the transfer probability matrix of the digital agriculture development level is computed using Matlab, considering a time span of one. This analysis aims to provide a more comprehensive understanding of the evolving trends in digital agriculture development. The results of this analysis are presented in Table 6.

**Table 6.** Matrix of transfer probabilities for the development level of digital agriculture.

| Province Status | Level of Digital Agriculture Development | | | |
|-----------------|-----------|------------------|-------------------|------------|
| | Low Level | Medium–Low Level | Medium–High Level | High Level |
| low level | 0.818 | 0.182 | 0.000 | 0.000 |
| Medium–low level | 0.015 | 0.803 | 0.182 | 0.000 |
| Medium–high level | 0.000 | 0.000 | 0.862 | 0.138 |
| high level | 0.000 | 0.000 | 0.000 | 1.000 |

Firstly, upon observing the main diagonal elements of the transfer probability matrix, it is evident that the probabilities surpass 0.800 and are consistently higher than the non-main diagonal elements. This indicates a higher probability for each province to maintain its current level of digital agriculture development in the subsequent period, implying a greater degree of stability in the development level. Notably, provinces with a high level of digital agriculture development exhibit a probability of 1.000 in maintaining their high level in the following year, signifying that their development level will not decline.

Secondly, by examining the non-primary diagonal elements of the transfer probability matrix, it is observed that the digital agriculture development level, apart from maintaining its current state, only transitions to neighboring categories. For instance, the probability of a province with a low development level of digital agriculture shifting to medium–low, medium–high, and high levels in the next year is 0.182, 0.000, and 0.000, respectively. This implies that provinces in the low level category primarily transition to the medium–low level while maintaining their original level. Provinces with a medium–low and medium–high level of digital agriculture development also exhibit similar characteristics. These

results can be attributed to the influence of various factors on the development of digital agriculture. Apart from economic development, natural resources and conditions play a significant role in shaping the trajectory of digital agriculture. Improving the development level of digital agriculture is a complex task that requires addressing the shortcomings of natural conditions and enhancing economic development. It is crucial for the government to formulate tailored plans for digital agriculture development based on the specific conditions of each province. Long-term strategies should be devised to foster the advancement of digital agriculture, reduce regional disparities, and promote sustainable agricultural development.

(2) Spatial transfer probability for the development level of digital agriculture.

Based on the results of the global Moran's I index and the local Moran's I index, it is evident that there is a positive spatial dependence and aggregation pattern in the development level of digital agriculture in China. To analyze the evolving trends in digital agriculture development more comprehensively, spatial factors were incorporated into the study. This involved examining the spatial Markov chain characteristics of the digital agriculture development level. Furthermore, utilizing the quartile principle, the digital agriculture development level was categorized into four types: low level, medium–low level, medium–high level, and high level. The transfer probability matrix of the digital agriculture development level was calculated, and the outcomes are presented in Table 7.

**Table 7.** Spatial transfer probability matrix for the development level of digital agriculture.

| Spatial Lagging | Province Status | Level of Digital Agriculture Development | | | |
|---|---|---|---|---|---|
| | | Low Level | Medium–Low Level | Medium–High Level | High Level |
| low level | low level | 0.857 | 0.143 | 0.000 | 0.000 |
| | medium–low level | 0.000 | 0.875 | 0.125 | 0.000 |
| | medium–high level | 0.000 | 0.000 | 0.714 | 0.286 |
| | high level | 0.000 | 0.000 | 0.000 | 1.000 |
| medium–low level | low level | 0.913 | 0.087 | 0.000 | 0.000 |
| | medium–low level | 0.000 | 0.917 | 0.083 | 0.000 |
| | medium–high level | 0.000 | 0.000 | 1.000 | 0.000 |
| | high level | 0.000 | 0.000 | 0.000 | 1.000 |
| medium-high level | low level | 0.632 | 0.368 | 0.000 | 0.000 |
| | medium–low level | 0.036 | 0.714 | 0.250 | 0.000 |
| | medium–high level | 0.000 | 0.000 | 0.792 | 0.208 |
| | high level | 0.000 | 0.000 | 0.000 | 1.000 |
| high level | low level | 0.900 | 0.100 | 0.000 | 0.000 |
| | medium–low level | 0.000 | 0.667 | 0.333 | 0.000 |
| | medium–high level | 0.000 | 0.000 | 0.938 | 0.063 |
| | high level | 0.000 | 0.000 | 0.000 | 1.000 |

Firstly, by examining the main diagonal elements of the transfer probability matrix, it is observed that the probability distributions are all greater than 0.632, surpassing the non-main diagonal elements. This suggests a stable digital agriculture development level, consistent with the findings of the traditional Markov chain analysis.

Secondly, for provinces with a high development level of digital agriculture, their future development level remains high, regardless of the spatial lag level.

Finally, the spatial lag level influences the probability of a province transferring its digital agriculture development level to some degree. In the case of provinces with a medium–low level, the type of spatial lag level increases the probability of transitioning to a medium–high level. For instance, the probabilities of transitioning to a medium–high level for provinces with a medium–low level of digital agriculture development are 0.125, 0.083, 0.025, and 0.333, respectively.

Based on the results of the spatial Markov chain analysis of digital agriculture development levels, it is evident that the development level remains stable, aligning with the findings of the traditional Markov chain analysis. However, it is worth noting that digital

agriculture development still exhibits a certain spatial effect, influencing the neighboring areas to some extent.

Overall, there is an obvious regional disparities in the development of digital agriculture in China. The differences in the eastern, central and western regions are relatively significant, especially the development level of digital agriculture in the western region lags behind.

## 4. Discussion and Conclusions

### 4.1. Discussion

In the existing literature, the measurement of the development level of digital agriculture mostly comes from the aspects of the environment, supporting resources, and outputs [44]. This study adds the level of agricultural informatization, as well as the digitization dimension of the agricultural production process, which measures the deep integration of digitalization and agriculture in the production process. On the basis of calculating the development level of digital agriculture and analyzing regional differences in previous studies [46,47], this paper uses the Markov chain to analyze the probability of horizontal transfer of digital agriculture and reveals that the development level of digital agriculture is relatively stable; therefore, improving digital agriculture is a challenge. This study further enriched the research on the regional development of digital agriculture and provided recommendations for improving the development level of digital agriculture and reducing regional disparities.

In addition, this study also addresses the following insights to reveal the reasons for regional disparities.

Firstly, the natural conditions and foundations for the development of agriculture are inequal in different regions. Considering that the development of agricultural industry is greatly constrained by the natural environment, therefore, it is more conducive to promote and use the large-scale machinery, as well as to apply the digital technology, in the agricultural production process in the areas with superior natural conditions and a solid foundation for the development of agriculture, such as Shandong, Heilongjiang and other places.

Secondly, there is a gap in China's digital economy, which leads to differences in the digital resources available in different regions, resulting in an unbalanced development of digital agriculture [39,41].

Thirdly, the level of infrastructure construction varies greatly. The application of digital technology and the promotion of mechanization need to rely on certain infrastructure, such as network construction and transportation facilities [48]. Due to the differences in economic level and geographical environment in different regions, the development of infrastructure construction is also extremely unbalanced, which is also one of the important reasons for regional disparities in digital agriculture.

Fourthly, the innovation and application of digital agriculture-related technologies is disparate. The level of innovation adoption and the scope and efficiency of the application of digital agriculture-related technologies are much higher in the eastern region than the western [49], which leads to the unbalanced development of digital agriculture in China.

### 4.2. Conclusions

Digital agriculture is a crucial direction for agricultural development, serving as a pivotal pathway to enhance sustainable agricultural practices. This study aims to assess the development level of digital agriculture in China by constructing an evaluation system comprising five dimensions: digital agriculture infrastructure, digital agriculture talent, agricultural informatization, the digitization of agricultural production processes, and agricultural production efficiency. Using data from 2013 to 2021, the objective sequential analysis method (SRA) is employed to assign weights to the indicators and calculate the evaluation values for the digital agriculture development level across the 31 provinces in China. Additionally, kernel density estimation, Moran's index, and Markov chain analysis

techniques are employed to examine the distribution pattern, spatial characteristics, and transition probabilities associated with the digital agriculture development level. The findings lead to the following conclusions:

(1) The overall development level of digital agriculture in the 31 provinces of China has been steadily improving; however, there are significant disparities exist between provinces and regions, with the western region lagging behind the eastern and central regions. Provinces with higher levels of digital agriculture development are concentrated in the eastern and central regions, with Sichuan being the exception in the west.

(2) The development level of digital agriculture shows less variation within the eastern and central regions of China, while significant disparities exist within the western region, leading to a polarization phenomenon. This situation poses a considerable disadvantage to sustainable agricultural development in the western region.

(3) Positive spatial dependence and aggregation characteristics are observed, indicating that provinces with similar digital agriculture development levels tend to cluster together. Within localized spatial areas, the disparities in digital agriculture development level among provinces are relatively small.

(4) The Markov chain analysis suggests that the digital agriculture development level tends to remain stable over time. Provinces are more likely to maintain their existing development level of digital agriculture in the following year. However, for provinces with a lower level of digital agriculture development, the type of spatial lag level somewhat increases the probability of their level transitioning.

*4.3. Recommendations*

The regional disparities in the development level of digital agriculture have emerged as a significant concern, presenting a formidable obstacle to the progress of digital agriculture and the attainment of sustainable agricultural development. It is important to recognize that enhancing the development level of digital agriculture is a gradual process influenced by a multitude of factors. In light of the aforementioned conclusions and the prevailing circumstances, this paper puts forth the following recommendations to effectively tackle the regional disparities in the development level of digital agriculture:

(1) Enhance policy formulation related to digital agriculture to eliminate regional disparities in its development. The government should devise policies based on the actual development conditions of each region, adjusting the distribution of digital technology resources nationwide. This will facilitate the inflow of digital resources into the western region, coordinate the overall development of digital agriculture nationwide, and elevate the level of digital agriculture development, thereby achieving sustainable agricultural development.

(2) Enhance infrastructure development and cultivate digital agriculture technical talents. Accelerate agricultural infrastructure construction and promote full network coverage in rural areas to provide a solid hardware foundation for the development of digital agriculture. Special attention should be given to addressing the inadequate infrastructure conditions in the western region.

(3) Investing in education and training programs at higher learning institutions and vocational colleges can create a skilled workforce capable of leveraging digital technologies in agriculture. This intellectual support is critical for successful digital adoption. Leverage the role of spatial radiation and regional assistance. Harnessing the spatial spillover effect from provinces with high digital agriculture development can be a catalyst for progress in neighboring provinces. To address the lower development level in the western region, it is recommended to implement inter-provincial assistance programs between the eastern, central, and western regions. This would involve establishing communication platforms and channels to facilitate knowledge exchange between provinces with high and low development levels of digital agriculture. These initiatives aim to enhance the digital agriculture development level in the

western region, ultimately achieving regional coordination and promoting sustainable development in digital agriculture.

(4)     Revealing the spatial dynamic evolution of digital agriculture development in China can provide valuable experience and inspiration for developing or emerging countries; meanwhile, it can also make a positive contribution to the sustainable development of global agriculture.

However, even though this study proposed to improve the development level of digital agriculture and eliminate the regional divide from the aspects of policy support, infrastructure, and personnel training, there are still many difficulties in the implementation of the above recommendations. For example, there will be different understandings regarding the implementation of policies related to digital agriculture in different regions. Therefore, the government should promote the balanced development of digital agriculture by interpreting the relevant policies of digital agriculture and strengthening the role of the government in infrastructure construction and personnel training.

### 4.4. Limitations and Future Directions

There are still some limitations of this study, which are also the directions of future research. First of all, in terms of index system construction, numerical indicators are currently selected. In the future, indicators created by mining text should be included; for example, in the "Government Work Report" analysis, the frequency of the words "digital agriculture" could be calculated, to further measure the importance of the digital agriculture governance. Secondly, the current measurement period is 2013–2021, which is a relatively short time span. This is due to the short development time of digital agriculture in China, and researchers are temporarily unable to use a longer period of continuous data for evaluation. Thirdly, take cities as samples to evaluate the development level of China's digital agriculture. Based on the current index system, some indicators cannot obtain city-level data. In the future, the development level of China's digital agriculture can be evaluated by constructing a city-level index system and enriching the methods of obtaining indicators.

**Author Contributions:** J.M. designed the study and revised the manuscript. B.Z. collected and analyzed data and wrote the draft. Y.S. collected related materials and wrote part of the draft. X.L. reviewed and edited the manuscript. All authors have read and agreed to the published version of the manuscript.

**Funding:** This research received no external funding.

**Institutional Review Board Statement:** Not applicable.

**Informed Consent Statement:** Not applicable.

**Data Availability Statement:** The data used in the study can be obtained from publicly available sources. The authors have provided the source of data in the article.

**Acknowledgments:** Special thanks to the reviewers for their valuable comments.

**Conflicts of Interest:** The authors declare no conflicts of interest.

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
