# Peer review of "Research on the Spatial Dynamic Evolution of Digital Agriculture—Evidence from China"

_sustainability, doi:10.3390/su16020735_

Round 1
Reviewer 1 Report
Comments and Suggestions for Authors
Based on data related to digital agriculture in various provinces and cities in China, this study explored the spatial trends and driving mechanisms of digital agriculture in China. There are some major concerns that the authors should address.
1. Lines 61-65,where are the sources of data such as China's agricultural production informatization level reaching 22.5% and the traceability rate of agricultural product quality and safety reaching 22.1%?
2. In the data, the authors should add more information about data, such as socio-economic data availability and access.
3. Is there any missing value in the data of each indicator in the evaluation index system constructed by this research institute?
4. If the data of the evaluation indicators can be accurate to the city level units, then the evaluation results will be more accurate.
5. A paragraph of limitation discussion should be added to clarify the limitation or uncertainty of data and methods in this current study.
6. More mechanism explanations are encouraged to further explain the spatial differences at the digital level and their underlying reasons.
7. Why did the authors set the research period to 2013-2021? Is the author's assessment of the spatial trend of digital agriculture based on 9 years of data inaccurate?
8. In order to further highlight the innovation of this article, it is better to compare the results of this study with other studies.
Author Response
Dear Editor and Reviewers:
On behalf of the co-authors, we are very grateful to you for giving us an opportunity to revise our manuscript. We really appreciate your valuable comments together with suggestions on our manuscript entitled "Research on the spatial dynamic evolution of digital agriculture -------- Evidence from China". We have therefore studied reviewers’ comments carefully and tried our best to revise our manuscript accordingly. The reviewer comments are laid out below in italicized font blue text and specific concerns have been numbered. Notably, our response is given in normal font and all changes/additions to the latest manuscript are using Microsoft Word’s Track changes method. Please see the attachment for a point-by-point response to the reviewers’ comments and concerns.
Thank you very much for your constructive review. Happy new year!

Reviewer 2 Report
Comments and Suggestions for Authors
Digital agriculture is a critical driver for advancing sustainable practices in the field, and this study focuses on evaluating the development of digital agriculture in China. The assessment is based on five key dimensions: digital agriculture infrastructure, digital agriculture talent, agricultural informatization, digitization of agricultural production processes, and agricultural production efficiency.
Using data spanning from 2013 to 2021, the study employs the objective sequential analysis method to assign weights to indicators and calculate evaluation values across the 31 provinces in China. Additionally, spatial analysis techniques such as kernel density estimation, Moran's index, and Markov chain analysis are utilized to examine distribution patterns, spatial characteristics, and transition probabilities related to digital agriculture development.
The findings reveal several important conclusions:
The overall development of digital agriculture in China is steadily improving, but significant regional disparities exist, with the western region lagging behind the eastern and central regions.
Disparities within the western region contribute to a polarization phenomenon, posing challenges to sustainable agricultural development.
Positive spatial dependence and aggregation characteristics indicate that provinces with similar digital agriculture development levels tend to cluster together.
Markov chain analysis suggests a tendency for the digital agriculture development level to remain stable over time, with provinces more likely to maintain their existing levels.
The study underscores the need to address regional disparities in digital agriculture development. Recommendations include enhancing policy formulation, infrastructure development, and talent cultivation. The paper emphasizes the importance of education and training programs, spatial radiation, and regional assistance to bridge the development gap. Ultimately, it highlights the significance of understanding the spatial dynamic evolution of digital agriculture in China as a valuable experience for global agriculture's sustainable development.
In conclusion, I agree with the publication of this work after small reviews.
Author Response
Dear Editor and Reviewers:
On behalf of the co-authors, we are very grateful to you for giving us an opportunity to revise our manuscript. We really appreciate your valuable comments together with suggestions on our manuscript entitled "Research on the spatial dynamic evolution of digital agriculture -------- Evidence from China". The authors are very grateful for your encouragement and support. We have further revised the manuscript under the suggestions of reviewers, and will continue to conduct the research of digital agriculture in the future. Please see the attachment.
Thank you very much for your constructive review. Happy new year!

Reviewer 3 Report
Comments and Suggestions for Authors
Abstract:
Provide more detailed explanations of how each of the five aspects mentioned in the index system contributes to the overall evaluation, enhancing clarity for readers.
Enhance the practical relevance of the study by adding a sentence or two on the implications of the findings and potential recommendations for policymakers or stakeholders.
Introduction:
Explain how the adoption of digital technologies contributes to the sustainability of agricultural practices, economic growth, and environmental conservation.
Provide context for the first three revolutions in agriculture to help readers better understand the historical progression leading to the consensus on "digital agriculture" as the fourth revolution.
Highlight China's significant role in global agriculture and digital agriculture, discussing how its experiences and developments can serve as a reference for other countries.
Conclude the introduction with a clear thesis statement outlining the main purpose of the paper, such as analyzing regional disparities in digital agriculture development in China and proposing actionable policy recommendations.
Ensure the introduction aligns with the most recent literature on digital agriculture, sustainable development, and regional disparities, mentioning any recent developments or trends to highlight the paper's relevance.
Results:
The results are well-explained.
Discussion:
Include a brief section discussing potential challenges or limitations in implementing the recommendations and propose solutions to address these challenges.
Improve the discussion section by incorporating recent literature that might be missing.
By addressing these points, the manuscript can be strengthened, providing a more comprehensive and insightful contribution to the field of digital agriculture in China.
Comments on the Quality of English LanguageMinor English editing is required
Author Response

(The authors gave the same response as above.)

Reviewer 4 Report
Comments and Suggestions for Authors
The authors did a wonderful job and is a well-written paper with lots of citations/supporting references. However, the organization of the paper, starting from Line 439 to the rest of the paper (Line 507) needs to be reorganized as you cannot have the discussion section after the concluding section. Honestly, I think the Discussion section should be the Recommendations of the paper. I expect the paper to be accepted for publication once my comments/concerns and those of other reviewers, if any, are fully addressed line-by-line (see my comments attached file). Good luck to the authors and looking forward to the revised copy of the manuscript.

English is fine, just need very minor changes as indicated in my report.
Author Response

(The authors gave the same response as above.)
